# Early combination of albumin with crystalloids administration might not be beneficial for the survival of ischemic stroke patients with sepsis: A retrospective analysis from MIMIC-IV database

Yuming Teng[1], Min Li[2]*, Mo Yuan[1]*

**1** School of Medicine, Yunnan University, Kunming, China, **2** Hospital Infection Management Office, First Affiliated Hospital of Kunming Medical University, Kunming, China

* 13888125580@163.com (ML); ym@ynu.edu.cn (MY)

## Abstract

The high incidence of sepsis following stroke is a key factor driving increased mortality. Despite the lack of clinical benefit from albumin therapy in stroke patients, emerging data support the use of crystalloid and albumin combination therapy for improving outcomes in sepsis. However, the potential utility of this combination therapy specifically in stroke patients with sepsis has yet to be determined. The objective of this study is to investigate the correlation between the combination of albumin with crystalloids and the 90-day mortality of ischemic stroke patients with sepsis. We included ischemic stroke patients with sepsis from the Medical Information Mart for Intensive Care (MIMIC)-IV database. Patients were categorized into the crystalloids group (crystalloids alone) and the combination group. Kaplan-Meier curves were applied to identify the survival probabilities of patients after different therapies for the 90-day. Multivariate COX regression analysis was used to investigate the association between different therapies and 90-days all-cause mortality. The Kaplan-Meier curve showed that patients in the combination group, especially those in the non-early combination group, had worse 90-day survival rate than those in the crystalloids group (all log-rank $P < 0.05$). In multivariate analyses, early combination therapy showed no significant association with reduced mortality, whereas non-early combination therapy remained independently associated with an increased risk of 90-day all-cause mortality (all $P < 0.05$). The above results were consistent in the post-PSM analyses and subgroup analyses. In ischemic stroke patients with sepsis, the administration of albumin combined with crystalloids was not associated with improved 90-day survival, regardless of whether the therapy was initiated early or not.

**Data availability statement:** All datasets supporting the conclusions of this study are obtained from the MIMIC-IV database (https://mimic.physionet.org/). The datasets can be obtained from github (https://github.com/neuroym/albumin-combination.git).

**Funding:** The author who received funding support is Mo Yuan, and this work was supported by grants of National Natural Science Foundation of China (No. 82300829). The fund's website is http://www.nsfc.gov.cn/. All funders in this study did not play any role in the study design, data collection and analysis, decision to publish, or preparation of the manuscript. They only provided necessary financial support to ensure the smooth progress of the research. We independently completed all stages of the research and ensured the objectivity and impartiality of the study.

**Competing interests:** The authors have declared that no competing interests exist.

## Introduction

Stroke is the world's second leading cause of death and the foremost cause of mortality in China [1]. Post-stroke infections are highly prevalent among stroke patients and may lead to sepsis [2,3]. Reported incidence rates of sepsis following stroke range from 2% to 13% [4–6], significantly contributing to the elevated mortality observed in these patients [7–9].

Sepsis is characterized by an inappropriate host response to infection, leading to organ dysfunction and substantial morbidity and mortality [10]. Fluid therapy is a critical component of sepsis treatment [11,12], significantly impacting patient outcomes, particularly in emergency settings [13]. For decades, there has been a contentious debate over the choice between using "crystalloids" versus "colloids" in the context of fluid therapy [10,14]. Due to their safety, cost-effectiveness, and widespread availability, crystalloids remain the primary choice for fluid administration in sepsis patients [10]. However, albumin, a type of colloid, remains longer in the intravascular space and more effectively increases volume due to its larger molecular weight [15]. Consequently, the combined use of crystalloids and albumin for initial resuscitation and subsequent volume replacement has become a widely adopted strategy and is recommended by the Surviving Sepsis Campaign (SSC) Guidelines [16]. Notably, recent studies have suggested that early combination therapy with crystalloids and albumin in patients with sepsis, septic shock, or cardiogenic shock may confer a survival advantage over crystalloid therapy alone in the intensive care unit (ICU) [17–19]. Therefore, despite ongoing debate, combined crystalloid and albumin therapy remains a viable option for treating sepsis in selected patients.

In contrast, clinical studies focused on acute ischemic stroke present a differing perspective on albumin therapy. Numerous experimental studies have demonstrated the neuroprotective properties of human albumin [20–23]. Major clinical trials have failed to demonstrate the neuroprotective benefits of albumin observed in experimental studies and have instead shown that early albumin administration may increase the risk of adverse effects, including pulmonary edema and cerebral hemorrhage [24,25].

This conflict between sepsis and stroke guidelines creates a clear therapeutic dilemma for the specific and clinically complex subgroup of ischemic stroke patients who develop sepsis in the ICU. It remains unclear whether the potential benefits of albumin for sepsis resuscitation outweigh its documented risks in the context of acute stroke, or how the timing of its administration might influence this balance.

Therefore, our study aims to determine whether combined crystalloid and albumin therapy improves survival in ischemic stroke patients with sepsis in the ICU compared to crystalloid therapy alone. We will also investigate the timing of combination therapy, comparing early and non-early administration to crystalloid therapy alone in the ICU. Given the lack of benefit and potential hazards associated with early albumin use in ischemic stroke patients [24,25], we hypothesize that the addition of albumin, whether early or not, will not enhance survival in ischemic stroke patients with sepsis.

## Materials and methods

### Database

The data for this study were extracted from the publicly accessible Medical Information Mart for Intensive Care IV (MIMIC-IV, version 2.2) database, which includes comprehensive clinical information on ICU patients admitted to the Beth Israel Deaconess Medical Center (BIDMC) from 2008 to 2019.

### Patient selection criteria

Patients diagnosed with cerebral infarction combined with sepsis (as defined by the sepsis-3 criteria) were included in the analysis. S1 Table in S1 File provides a list of the International Classification of Diseases (ICD)-9 and ICD-10 codes that are used to diagnose cerebral infarction. The exclusion criteria were: (1) patients under 18 years old at admission; (2) readmission to the hospital; (3) non-first ICU admission; and (4) patients who had not received any crystalloids administration. A total of 493 patients who had received either combination administration of albumin and crystalloids or crystalloids alone were included in this study.

### Ethics statement

MIMIC-IV is a public clinical database co-developed by Beth Israel Deaconess Medical Center (BIDMC) and MIT, sourced from BIDMC's deidentified routine care data. Its creation and public sharing received an ethical waiver from BIDMC's Institutional Review Board, which granted informed consent exemption and approved data sharing [26]. Access is restricted to investigators who have completed required ethics training and have signed a Data Use Agreement [27]. In compliance with these requirements, our research team has fulfilled all access prerequisites. Therefore, this retrospective study meets the exemption criteria for ethical review,and this practice has an established precedent in peer-reviewed studies utilizing the MIMIC-IV database [18,28].

### Data extraction

The data of all patients were extracted from the MIMIC-IV database by using Structured Query Language (SQL) with PostgreSQL software. The following parameters was obtained: (1) demographic information: age, and gender; (2) vital signs: mean arterial blood pressure (MAP), respiratory rate, heart rate, and percutaneous oxygen saturation ($SpO_2$); (3) comorbidities: hypertension, congestive heart failure, atrial fibrillation, diabetes, respiratory failure, chronic kidney disease, chronic pulmonary disease, and hyperlipidemia; (4) laboratory indicators: peripheral white blood cell (WBC), blood urea nitrogen (BUN), serum creatinine, platelet, hemoglobin, and lactate; (5) severity scores: Sequential Organ Failure Assessment (SOFA), Glasgow Coma Scale (GCS), and Acute Physiology Score III (APS III); (6) clinical treatment: ventilation, renal replacement therapy (RRT), vasopressor use, recombinant tissue plasminogen activator (RTPA) therapy, endovascular treatment, antibiotic therapy, albumin administration, and crystalloids administration.

### Time to combined

The timing of albumin combined is defined as the time of albumin administration minus the time of crystal administration. In patients, receiving albumin combined within the first 24 hours after crystalloids administration is defined as the early combination group (n = 120), while those who do not receive albumin combination therapy during this period are defined as the non-early combination group (n = 75).

### Primary and secondary outcomes

The primary endpoint event was 90-day all-cause mortality. Secondary endpoint events included length of hospital stay (LOS hospital) and length of ICU stay (LOS ICU).

## Statistical analysis

After the normality test, continuous variables were expressed as median (interquartile range) and compared across groups using the Mann-Whitney U test or Kruskal–Wallis H test. Categorical variables were expressed as n (%) and compared across groups using the chi-square test or Fisher's exact test.

To minimize potential bias between the crystalloids alone group and the combination group, propensity score matching (PSM) was performed. The following variables were matched: age, gender, SOFA score, GCS score, and APS III score. The matching process was conducted with a caliper width of 0.25 of the pooled standard deviation of the logit of the propensity score. This matching was performed at a 1:1 based on baseline characteristics to ensure comparability between the groups.

Kaplan-Meier curves and log-rank tests were applied to identify the survival probabilities of patients after different therapies for the 90-day. Cox proportional hazards regression models were used to investigate the association between different therapies and 90-day all-cause mortality, with the corresponding hazard ratio (HR) with 95% confidential interval (CI) calculated. To examine time-varying treatment effects, we also performed a time-dependent Cox regression analysis, segmenting the follow-up period into early (0–28 days) and late (29–90 days) phases. Additionally, a multivariable Gamma regression model was applied to explore the association between different therapies and two outcomes: LOS hospital and LOS ICU. Following a univariate analysis, variables with a $P$-value of less than 0.05, as well as clinically relevant variables identified as potential risk factors, were selected for the multivariable analysis. We constructed three multivariable models. Model 1 was adjusted for age and sex. In addition to the variables in Model 1, Model 2 was adjusted for lactate, APS III score, and WBC count. Additionally, Model 3 was adjusted for all variables in Model 2 plus vasopressor use, timing of antibiotic administration, congestive heart failure, chronic kidney disease, chronic pulmonary disease, and glycopeptide use. Finally, we also performed a subgroup analysis according to age, gender, MAP, chronic pulmonary disease, chronic kidney disease, congestive heart failure, RRT, lactate, SOFA, vasopressor, timing of antibiotic administration, and timing of crystalloid administration. All analyses were conducted using R version 4.4.3. Two-side $P$-values < 0.05 were considered statistically significant.

## Results

### Clinical characteristics

The process of patient selection and exclusion is shown in Fig 1, and 493 patients fulfilled all screening criteria and were enrolled in our study. Among these patients, 298 patients had used crystalloids alone (crystalloids group), and 195 patients received albumin combined with crystalloids (combination group).

The detailed general data of the patients are shown in Table 1. The median age of the included patients is 66.94 years (IQR: 56.12–76.48 years), and 43.61% were female. Compared with the crystalloids group, patients in the combination group were more likely to report comorbidities of congestive heart failure, atrial fibrillation, respiratory failure, chronic kidney disease, and chronic pulmonary disease. In addition, they also had more adverse laboratory parameters (elevated BUN, serum creatinine, and lactate, and lower hemoglobin), higher illness severity scores (SOFA, GCS, and APS III; all $P$ < 0.001) on the first ICU day, and greater requirements for mechanical ventilation and RRT. In the combination group, 20.51% received vasopressors within 24 hours of crystalloid initiation, and 9.23% received them after 24 hours. In contrast, the vast majority (92.95%) in the crystalloids group did not require vasopressors, with only 3.02% and 4.03% receiving early and late administration, respectively. Compared to the crystalloids group, the combination group employed glycopeptides (80.51% vs. 66.78%, $P$ < 0.001) and carbapenems (12.82% vs. 7.38%, $P$ = 0.044) more frequently, and underwent earlier antibiotic administration (55.90% vs. 43.29%, $P$ = 0.006). Finally, despite a significantly higher median crystalloid volume in the combination group ($P$ < 0.001), the timing of the initial crystalloid administration was similar between groups.

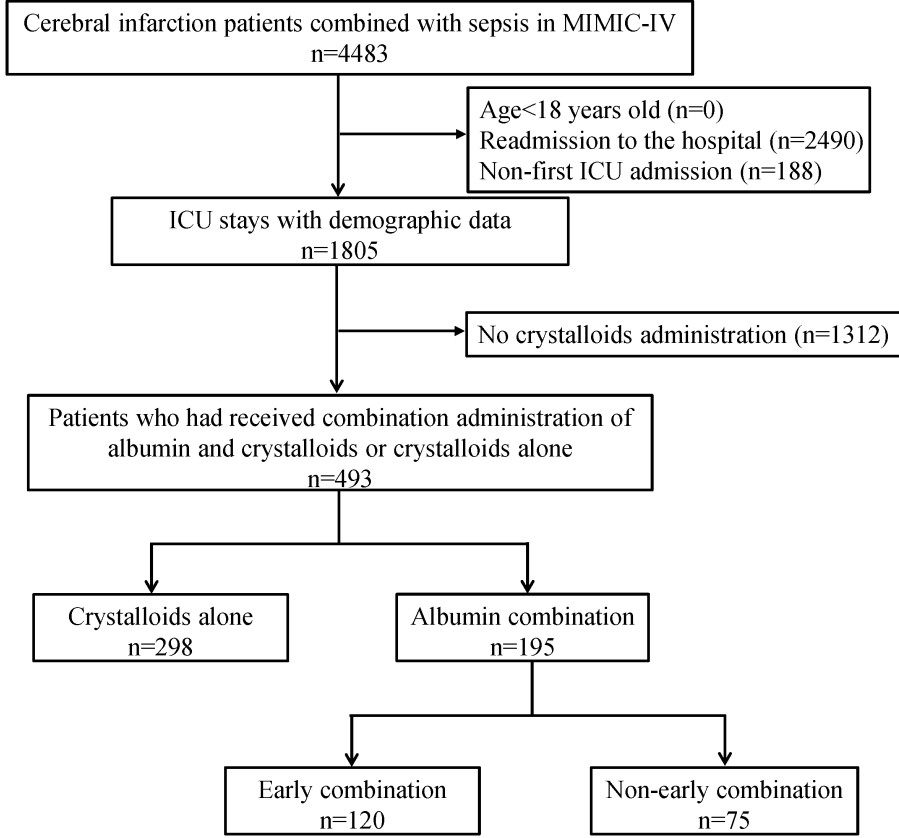

**Fig 1. Schematic diagram of study sample selection steps.** MIMIC, Medical Information Mart for Intensive Care; ICU, intensive care unit.

Following 1:1 propensity score matching, 160 patients from the combination group were matched to 160 from the crystalloids group. The baseline characteristics of the matched population were comparable, and specifically, no statistically significant differences remained in comorbidities or severity scores between the groups (S2 Table in S1 File).

**Primary outcome**

The Kaplan-Meier curve showed that patients received combination therapy had significantly higher 90-day all-cause mortality than those received crystalloids alone (combination group: 80.3% vs. crystalloids group: 61.2%, log-rank $P = 0.004$) (Fig 2A). Moreover, further analysis revealed not only a lack of survival benefit in the early combination group, but also a significantly higher mortality rate in the non-early combination group (93.8%) compared to both the crystalloids (61.2%) and early combination (70.5%) groups (log-rank $P < 0.001$) (Fig 2B). Similarly, in the matched cohort, the patients in the combination group, especially those in the non-early combination group, had higher 90-day all-cause mortality than those in the crystalloids group (Fig 2C, D).

In univariable Cox regression analysis, patients received non-early combination therapy had a significantly higher risk of 90-day all-cause mortality than those received crystalloids alone. This association remained consistent both before and after PSM (S3, S4 Tables in S1 File). Subsequently, variables identified as potential risk factors and those with $P$-values <0.05 in the univariable analysis were included in a multivariable Cox regression model. After adjustment in Model 1, combination therapy was significantly associated with an increased risk of 90-day all-cause mortality (HR 1.65, 95% CI

**Table 1. Baseline characteristics between the two groups.**

| Categories | Total population n=493 | Crystalloids alone n=298 | Combination n=195 | P value |
|---|---|---|---|---|
| Demographic | | | | |
| Age (years) (median, IQR) | 66.94; 56.12-76.48 | 65.71; 54.17-76.33 | 67.98; 59.50-77.67 | 0.043 |
| Female, n (%) | 215 (43.61) | 125 (41.95) | 90 (46.15) | 0.357 |
| Vital signs (median, IQR) | | | | |
| MAP (mmHg) | 80.15; 73.04-88.85 | 83.37; 75.98-91.38 | 75.16; 70.83-81.56 | <0.001 |
| Respiratory rate (breaths/min) | 18.67; 16.54-20.85 | 19.00; 16.89-20.86 | 17.98; 15.98-20.76 | 0.014 |
| Heart rate (beats/min) | 81.74; 72.84-91.81 | 82.39; 72.36-91.95 | 81.42; 73.27-91.32 | 0.540 |
| SpO2 (%) | 98.00; 96.71-98.96 | 97.98; 96.67-98.94 | 98.16; 96.83-98.96 | 0.524 |
| Comorbidities, n (%) | | | | |
| Hypertension | 368 (74.65) | 217 (72.82) | 151 (77.44) | 0.249 |
| Congestive heart failure | 131 (26.57) | 65 (21.81) | 66 (33.85) | 0.003 |
| Atrial fibrillation | 190 (38.54) | 96 (32.21) | 94 (48.21) | <0.001 |
| Diabetes | 163 (33.06) | 94 (31.54) | 69 (35.38) | 0.375 |
| Respiratory failure | 246 (49.90) | 138 (46.31) | 108 (55.38) | 0.049 |
| Chronic kidney disease | 92 (18.66) | 40 (13.42) | 52 (26.67) | <0.001 |
| Chronic pulmonary disease | 114 (23.12) | 57 (19.13) | 57 (29.23) | 0.009 |
| Hyperlipidemia | 235 (47.67) | 143 (47.99) | 92 (47.18) | 0.861 |
| Laboratory parameters (median, IQR) | | | | |
| WBC ($10^9$/L) | 14.50; 10.70-18.60 | 14.20; 10.50-18.30 | 14.90; 11.20-19.60 | 0.053 |
| BUN (mg/dL) | 19.00; 14.00-27.00 | 17.00; 13.00-25.00 | 21.00; 16.00-29.00 | 0.001 |
| Serum creatinine (mg/dL) | 1.00; 0.80-1.40 | 1.00; 0.80-1.20 | 1.10; 0.80-1.70 | 0.002 |
| Platelet ($10^9$/L) | 213.00; 166.00-276.50 | 225.00; 182.00-285.00 | 195.00; 150.00-258.00 | 0.094 |
| Hemoglobin (g/dL) | 11.70; 10.30-13.50 | 12.20; 10.50-13.70 | 11.30; 10.00-12.80 | <0.001 |
| Lactate (mmol/L) | 2.90; 1.90-4.80 | 2.40; 1.70-3.80 | 3.80; 2.60-6.80 | <0.001 |
| Severity scores (median, IQR) | | | | |
| SOFA | 3; 2-4 | 3; 2-4 | 3; 2-5 | <0.001 |
| GCS | 15; 11-15 | 14; 10-15 | 15; 14-15 | <0.001 |
| APS III | 42; 31-58 | 38; 29-52 | 49; 36-65 | <0.001 |
| Treatment, n (%) | | | | |
| Ventilation | 473 (95.94) | 280 (93.96) | 193 (98.97) | 0.006 |
| RRT | 43 (8.72) | 12 (4.03) | 31 (15.90) | <0.001 |
| Vasopressor[a] | | | | <0.001 |
| None | 414 (83.98) | 277 (92.95) | 137 (70.26) | |
| Within 24 hours | 49 (9.94) | 9 (3.02) | 40 (20.51) | |
| After 24 hours | 30 (6.09) | 12 (4.03) | 18 (9.23) | |
| RTPA | 3 (0.61) | 1 (0.34) | 2 (1.03) | 0.335 |
| Endovascular treatment | 43 (8.72) | 31 (10.40) | 12 (6.15) | 0.102 |
| Antibiotic-Carbapenems | 47 (9.53) | 22 (7.38) | 25 (12.82) | 0.044 |
| Antibiotic- Glycopeptide | 356 (72.21) | 199 (66.78) | 157 (80.51) | <0.001 |
| Antibiotic-β-lactams | 416 (84.38) | 249 (83.56) | 167 (85.64) | 0.533 |
| Antibiotic-Aminoglycosides | 39 (7.91) | 24 (8.05) | 15 (7.69) | 0.884 |
| Timing of antibiotic administration[b] | | | | 0.006 |
| Within 24 hours | 238 (48.28) | 129 (43.29) | 109 (55.90) | |
| After 24 hours | 255 (51.72) | 169 (56.71) | 86 (44.10) | |

*(Continued)*

**Table 1.** (Continued)

| Categories | Total population n=493 | Crystalloids alone n=298 | Combination n=195 | *P* value |
|---|---|---|---|---|
| Timing of crystalloid administration[c] (hours) (median, IQR) | 5.43; 2.06-28.69 | 7.02; 1.84-42.11 | 5.00; 2.53-14.73 | 0.228 |
| Volume of crystalloid administration (mL) (median, IQR) | 1750; 1000-3000 | 1300; 800-2163 | 2700; 1500-3800 | <0.001 |

IQR, interquartile range; MAP, mean arterial blood pressure; WBC, white blood cell; BUN, blood urea nitrogen; SOFA, sequential organ failure assessment; GCS, Glasgow coma scale; APS III, acute physiology score-III; RRT, renal replacement therapy; RTPA, recombinant tissue plasminogen activator.

[a]The timing of vasopressor use was defined as the time interval from the initiation of crystalloid infusion to the first administration of a vasopressor. Based on this, patients were categorized into three groups: no vasopressor use, early use (within 24 hours of crystalloid initiation), and late use (after 24 hours).

[b]The timing of antibiotic administration was defined as the interval from ICU admission to the first dose of antibiotics. Patients were accordingly categorized into two groups: those who received antibiotics within 24 hours of ICU admission, and those who received them after 24 hours.

[c]The timing of crystalloid administration was defined as the interval from ICU admission to the first administration of crystalloid fluid.

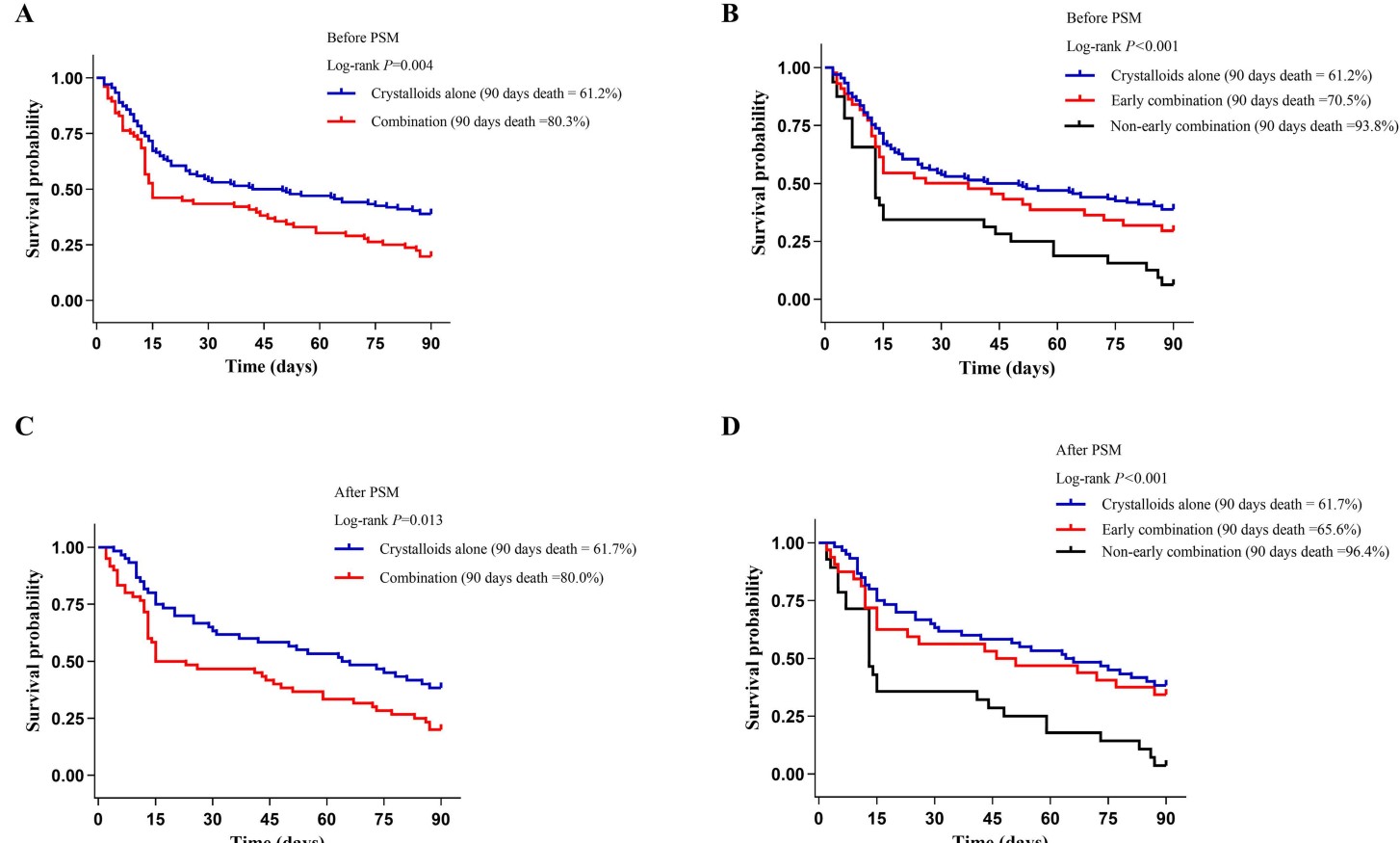

**Fig 2. Kaplan–Meier survival analysis curves for 90-days all-cause mortality.** Kaplan-Meier survival analysis was performed to compare the survival probabilities between the crystalloids group and the combination group, both (A) before and (C) after 1:1 propensity score matching; Kaplan-Meier survival analysis was performed to compare the crystalloids group, the early combination group, and the non-early combination group, both (B) before and (D) after 1:1 propensity score matching.

1.18–2.30, *P* = 0.003), and this association remained significant in Models 2 (HR 1.56, 95% CI 1.08–2.24, *P* = 0.017) and 3 (HR 1.59, 95% CI 1.09–2.34, *P* = 0.017). Further analysis revealed that non-early combination therapy was associated with an even higher risk of 90-day all-cause mortality across all models (Model 1: HR 2.30, 95% CI 1.51–3.51, *P* < 0.001; Model 2: HR 2.20, 95% CI 1.40–3.48, *P* < 0.001; Model 3: HR 2.38, 95% CI 1.46–3.88, *P* < 0.001) (Table 2). Notably, early combination therapy demonstrated no significant association with reduced mortality in any model. These findings were consistent in the post-PSM cohort (S5 Table in S1 File). In addition, time-dependent Cox regression analysis revealed no significant differences in the risk of 90-day all-cause mortality in the early phase between the crystalloids group and either combination subgroup. However, in the late phase, a substantially higher risk of 90-day all-cause mortality was observed for the non-early combination group, with the association remaining robust and notably strengthening in the adjusted model (HR = 6.18, 95% CI 2.86–13.35, *P* < 0.001) (S6 Table in S1 File). The consistency of these findings was confirmed in the post-PSM cohort (S7 Table in S1 File).

### Secondary outcomes

Patients receiving combination therapy had significantly longer durations of both hospital stay (20.96 (12.20–32.74) days vs. 16.05 (9.03–26.02) days; *P* = 0.001) and ICU stay (11.96 (5.29–21.04) days vs. 8.25 (3.94–14.99) days; *P* < 0.001) than those receiving crystalloids alone (Fig 3A1, 3C1). In addition, there was no significant difference in the length of hospital stay and ICU stay between the early combination group and the crystalloids group, but prolonged stays in hospital (22.58. (14.69–34.16) versus 16.05 (9.03–26.02) days, *P* < 0.001) and ICU (15.54 (9.46–24.01) versus 8.25 (3.94–14.99) days, *P* < 0.001) were significantly more likely in the non-early combination group (Fig 3B1, 3D1). After PSM, the results are obtained to be similar (Fig 3A2–3D2).

After adjustment across multiple models, combination therapy, particularly non-early combination therapy, remained significantly associated with prolonged durations of both hospital and ICU stays (Table 2). In contrast, early combination

**Table 2. Multivariable analysis of the association between therapies and outcomes.**

| Outcomes | Crystalloids | Combination | *P* value | Crystalloids | Early combination | Non-early combination | *P* value[a] | *P* value[b] |
|---|---|---|---|---|---|---|---|---|
| 90-day mortality | | | | | | | | |
| Unadjusted | 1.00 (reference) | 1.64 (1.18-2.29) | 0.004 | 1.00 (reference) | 1.28 (0.84-1.93) | 2.34 (1.54-3.57) | 0.246 | <0.001 |
| Model 1 | 1.00 (reference) | 1.65 (1.18-2.30) | 0.003 | 1.00 (reference) | 1.29 (0.85-1.96) | 2.30 (1.51-3.51) | 0.228 | <0.001 |
| Model 2 | 1.00 (reference) | 1.56 (1.08-2.24) | 0.017 | 1.00 (reference) | 1.22 (0.78-1.89) | 2.20 (1.40-3.48) | 0.383 | <0.001 |
| Model 3 | 1.00 (reference) | 1.59 (1.09-2.34) | 0.017 | 1.00 (reference) | 1.24 (0.79-1.96) | 2.38 (1.46-3.88) | 0.344 | <0.001 |
| LOS Hospital | | | | | | | | |
| Unadjusted | 1.00 (reference) | 1.21 (1.06-1.40) | 0.006 | 1.00 (reference) | 1.15 (0.97-1.35) | 1.32 (1.09-1.62) | 0.105 | 0.005 |
| Model 1 | 1.00 (reference) | 1.27 (1.11-1.45) | <0.001 | 1.00 (reference) | 1.22 (1.04-1.43) | 1.35 (1.12-1.63) | 0.014 | 0.002 |
| Model 2 | 1.00 (reference) | 1.16 (1.01-1.34) | 0.038 | 1.00 (reference) | 1.10 (0.93-1.31) | 1.25 (1.03-1.51) | 0.246 | 0.024 |
| Model 3 | 1.00 (reference) | 1.13 (0.98-1.29) | 0.077 | 1.00 (reference) | 1.07 (0.92-1.26) | 1.21 (1.02-1.45) | 0.374 | 0.038 |
| LOS ICU | | | | | | | | |
| Unadjusted | 1.00 (reference) | 1.36 (1.16-1.61) | <0.001 | 1.00 (reference) | 1.20 (0.99-1.47) | 1.62 (1.29-2.06) | 0.063 | <0.001 |
| Model 1 | 1.00 (reference) | 1.41 (1.20-1.67) | <0.001 | 1.00 (reference) | 1.27 (1.04-1.55) | 1.63 (1.30-2.07) | 0.018 | <0.001 |
| Model 2 | 1.00 (reference) | 1.29 (1.08-1.54) | 0.004 | 1.00 (reference) | 1.13 (0.92-1.40) | 1.50 (1.19-1.90) | 0.221 | <0.001 |
| Model 3 | 1.00 (reference) | 1.18 (1.00-1.41) | 0.049 | 1.00 (reference) | 1.04 (0.85-1.28) | 1.37 (1.11-1.72) | 0.680 | 0.005 |

Data are presented as the HR (95%CI). Model 1 was adjusted for age and sex. Model 2: Model 1 plus lactate, APS III score, and WBC. Model 3: Model 2 plus vasopressor use, timing of antibiotic administration, congestive heart failure, chronic kidney disease, chronic pulmonary disease,and glycopeptide use.

*P* value[a] was used to indicate the differences between the crystalloids group and the early combination group. *P* value[b] was used to indicate the differences between the crystalloids group and the non-early combination group.

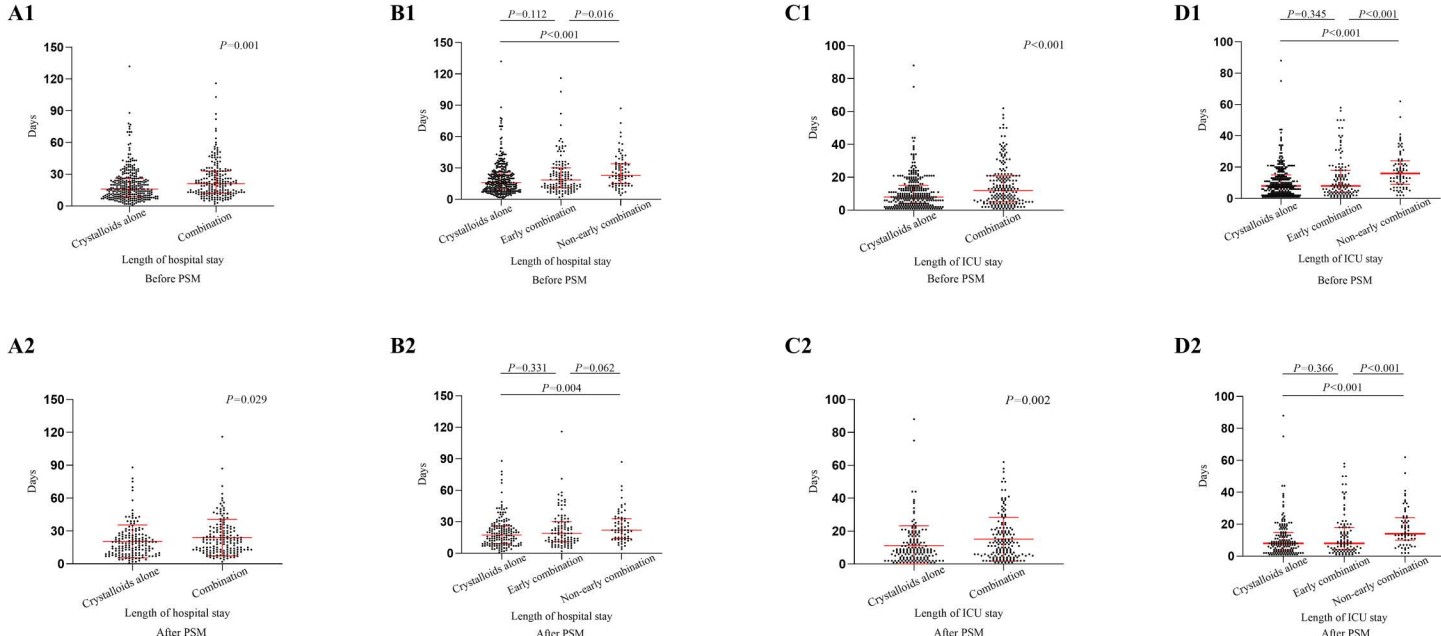

**Fig 3. Length of stay in hospital and ICU.** Length of stay in hospital compared between the crystalloids group and the combination group (A1) before and (A2) after propensity score matching (1:1); Length of stay in hospital among the crystalloids group, the early combination group, and the non-early combination group (B1) before and (B2) after propensity score matching (1:1); Length of stay in ICU compared between the crystalloids group and the combination group (C1) before and (C2) after propensity score matching (1:1); Length of stay in ICU among the crystalloids group, the early combination group, and the non-early combination group (D1) before and (D2) after propensity score matching (1:1).

therapy showed no significant reduction in the length of hospitalization or ICU care, whether before or after PSM (Table 2, S5 Table in S1 File).

## Subgroup analyses

We performed subgroup and stratified analyses to evaluate the association between combination therapy and 90-day all-cause mortality. Tests for interaction showed no statistical significance for age, gender, chronic pulmonary disease, chronic kidney disease, congestive heart failure, RRT, lactate level, SOFA score, timing of vasopressor use, timing of anti-biotic administration, or timing of crystalloid administration (interaction $P$-values ranging from 0.087 to 0.953), both before and after PSM (Fig 4, S1 Fig in S1 File). However, interaction tests indicated that higher MAP was associated with better outcomes only in the early combination therapy group (interaction $P=0.042$), although this finding was not significant in the PSM analysis (interaction $P=0.210$) (Fig 4, S1 Fig in S1 File).

## Discussion

Our study found that among ischemic stroke patients with sepsis, those who received combined crystalloid and albumin therapy had a higher 90-day mortality rate and experienced longer ICU and hospital lengths of stay. Further analysis showed that, compared to crystalloid therapy alone, neither early nor non-early combination therapy was associated with improved 90-day survival or reduced duration of hospitalization. In contrast, non-early combination therapy was associated with poorer 90-day survival, particularly beyond 28 days, as well as prolonged ICU and hospital stays.

Both sepsis and ischemic stroke share common pathophysiological features, including hypoperfusion, oxidative stress injury, endothelial dysfunction, and coagulation abnormalities. Albumin, beyond its role in volume expansion, provides

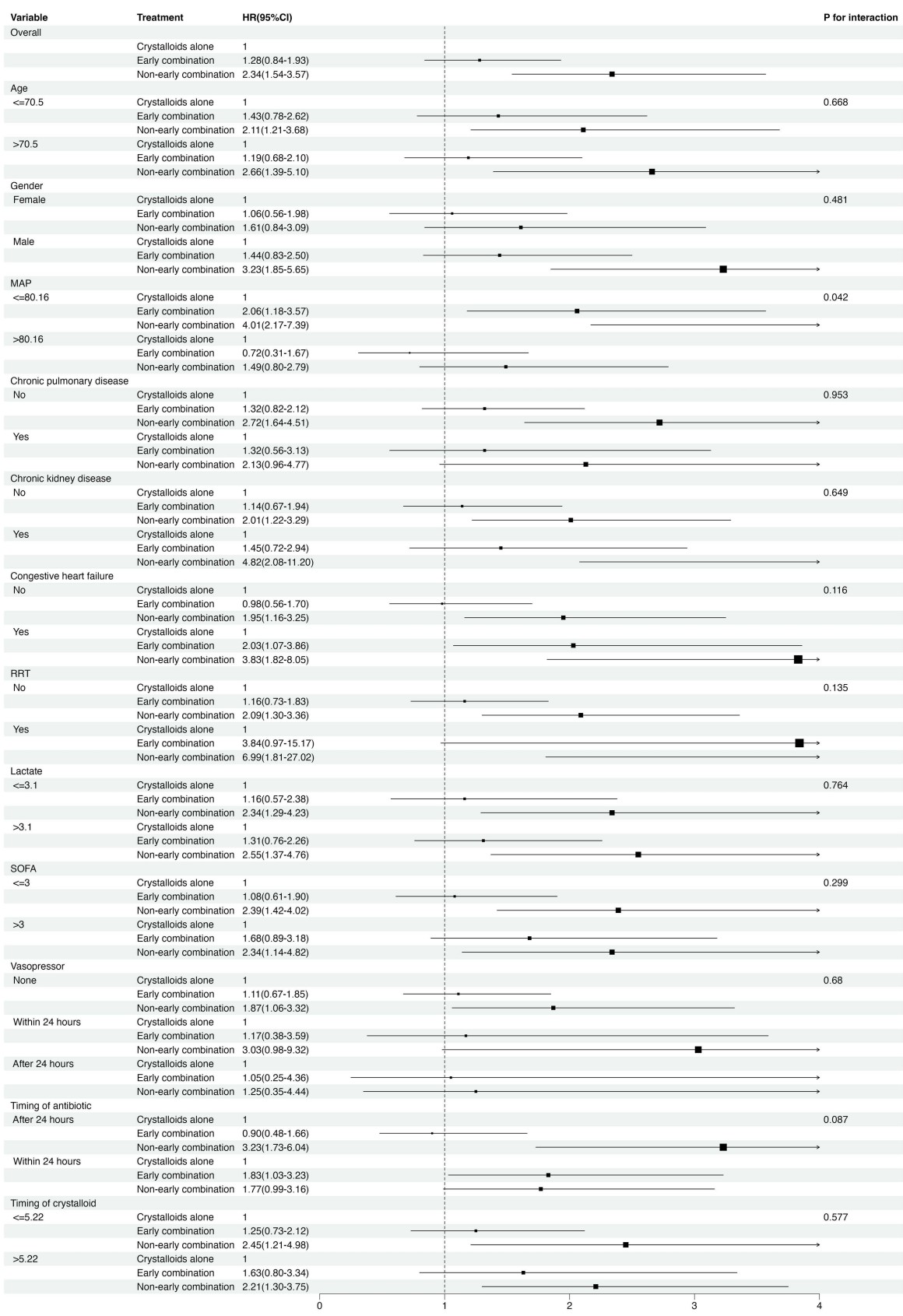

**Fig 4. Forest plot for subgroup analysis of the relationship between combination therapy and 90-days all-cause mortality.** HR, hazard ratio; 95% CI, 95% confidence interval; MAP, mean arterial blood pressure; RRT, renal replacement therapy; SOFA, sequential organ failure assessment.

essential biological functions such as, exerting antithrombotic and antioxidant effects, and supporting vascular integrity and molecular transport [15,29,30]. The use of combined crystalloid and albumin for fluid resuscitation in sepsis remains highly controversial. Most previous studies have not shown a clear survival benefit in general sepsis populations [31,32], although some data suggest potential improvements in endothelial function [29] and even a modest 90-day survival advantage in patients with septic shock (HR = 0.87; 95% CI: 0.77–0.99) [32]. Reflecting this mixed evidence, the latest SSC sepsis guidelines offer a weak recommendation for albumin supplementation in patients requiring large-volume crystalloid resuscitation [16].

Notably, recent investigations have highlighted the critical importance of timing. A pilot study demonstrated improved skin endothelial function when albumin was administered within 24 hours of septic shock onset [29]. Furthermore, early combination therapy (albumin given after crystalloid within the first 24 hours) has been associated with improved survival rates in sepsis patients, whereas non-early combination beyond 24 hours showed no such benefit, with survival advantages potentially confined to the first 28 days [18]. Subsequent studies across various sepsis/septic shock patient populations with different comorbidities/etiology have corroborated that combination therapy within 24 hours can improve outcomes or reduce ICU and hospital stays [17–19]. Our findings are consistent with previous research in that non-early combination therapy provided no benefit and was instead associated with poor 90-day outcomes, particularly negatively impacting survival beyond 28 days. The inconsistency lies in the fact that early combination therapy did not improve outcomes or shorten ICU and hospital stays. From the pathophysiological perspective of sepsis, our findings may be related to the progression of vascular endothelial injury—a hallmark feature of sepsis that intensifies and peaks within the first 24 hours [33–35]. Vascular endothelial injury causes albumin to leak into the interstitial space, and albumin infusion can exacerbate this extravasation [35]. Therefore, non-early albumin administration occurs after maximal endothelial dysfunction has developed. At this stage, albumin infusion may intensify extravasation, potentially exacerbating tissue edema, impairing volume expansion, and ultimately leading to clinical deterioration, prolonged ICU stays, and increased mortality. The lack of benefit from early combination therapy, however, warrants further investigation by considering albumin's potential effects in the context of ischemic stroke.

For ischemic stroke, beyond reperfusion therapy, neuroprotection represents a key therapeutic strategy. The pathophysiological processes of stroke, including hypoperfusion, oxidative stress, endothelial dysfunction, and coagulation abnormalities, contribute to neural injury. Albumin's potential to ameliorate these processes has positioned it as a candidate neuroprotective agent. Consequently, numerous studies have explored albumin as a neuroprotectant in ischemic stroke. Previous animal studies demonstrated that albumin reduces infarct volume [21], improves neurological function scores [22], and reverses DWI changes [23], suggesting neuroprotective effects with a therapeutic window of up to 72 hours [20]. Subsequent clinical studies initially suggested that early albumin administration (within 5 hours) could replicate these benefits [36,37]. However, numerous clinical trials and meta-analyses have not confirmed that early albumin therapy improves neurological function or reduces 90-day and 180-day mortality rates in ischemic stroke patients [24,25,38,39]. This may be attributed to potential adverse effects such as pulmonary edema and cerebral hemorrhage, linked to albumin's antiplatelet effects, hemodilution of coagulation factors, and calcium-binding properties [24]. Although studies on non-early albumin combination therapy in isolated ischemic stroke are lacking, these findings are consistent with our observation that early albumin combination therapy was not associated with improved 90-day mortality or shorter ICU and hospital stays in septic patients with ischemic stroke.

To the best of our knowledge, our study is the first to investigate the correlation between early albumin-crystalloid combination therapy and 90-day mortality in ischemic stroke patients with sepsis in the ICU. Our findings suggest that

ischemic stroke with sepsis patients, the potential benefits of early albumin combination therapy—possibly including superior volume expansion, endothelial stabilization, and neuroprotection—may be counterbalanced by its associated risks of pulmonary edema and cerebral hemorrhage. In the non-early phase, severe endothelial damage likely facilitates enhanced albumin extravasation, potentially exacerbating systemic (including cerebral) edema and hemorrhage. These complications could lead to more complex or deteriorating clinical conditions, thereby potentially prolonging hospitalization and increasing 90-day mortality, particularly beyond 28 days. The fact that non-early combination therapy was specifically associated with late (post-28-day) outcomes, rather than early mortality, suggests that this association may not be solely driven by baseline disease severity, as severity would theoretically exert its strongest effect on early survival. Taken together, our study suggests that the use of combined crystalloid and albumin therapy was not associated with improved outcomes in ischemic stroke patients with sepsis, regardless of timing, when compared to crystalloids alone.

Our study has several limitations. First, while propensity score matching was used to minimize confounding, it significantly reduced the sample size. Second, despite rigorous matching, unmeasured residual confounders may persist. Third, the absence of data on stroke severity (e.g., NIHSS scores), infarct and penumbra volume prevented adjustment for these potentially influential variables. Fourth, it is important to note that the pathophysiological explanations for the potential harm of late albumin administration remain hypothetical. While they are grounded in established pathophysiology and supported by literature, they were not directly validated by our study. Thus, they remain plausible hypotheses requiring further validation. Fifth, as a retrospective analysis, the level of evidence is limited, and future randomized controlled trials are needed to confirm our findings.

## Conclusion

Overall, our retrospective study indicates that the combination of albumin with crystalloids did not improve 90-day survival, regardless of timing. Analysis based on timing showed that early combination therapy provided no significant benefit, while non-early therapy was associated with worse outcomes. Future prospective randomized trials are needed to confirm these findings.

## Supporting information

**S1 File. S1 Table.** ICD-9 and ICD-10 codes for cerebral infarction. **S2 Table.** Baseline characteristics between the two groups after propensity score matching. **S3 Table.** Univariable COX analysis of the association between therapies and 90-day all-cause mortality before propensity score matching. **S4 Table.** Univariable COX analysis of the association between therapies and 90-day all-cause mortality after propensity score matching. **S5 Table**. Multivariable analysis of the association between therapies and outcomes after propensity score matching. **S6 Table**. Time-dependent Cox regression analysis of the association between therapies and outcomes before propensity score matching. **S7 Table**. Time-dependent Cox regression analysis of the association between therapies and outcomes after propensity score matching. **S1 Fig.** Forest plot for subgroup analysis of the relationship between combination therapy and 90-day all-cause mortality after propensity score matching.
(DOCX)

## Acknowledgments

The authors express their deep gratitude to all the researchers who built and maintained the MIMIC IV database.

## Author contributions

**Conceptualization:** Yuming Teng.

**Data curation:** Mo Yuan.

**Formal analysis:** Mo Yuan.

**Funding acquisition:** Mo Yuan.

**Investigation:** Mo Yuan, Yuming Teng.

**Methodology:** Mo Yuan, Yuming Teng.

**Project administration:** Mo Yuan, Yuming Teng.

**Resources:** Mo Yuan, Yuming Teng.

**Software:** Mo Yuan, Yuming Teng.

**Supervision:** Yuming Teng, Min Li.

**Validation:** Mo Yuan, Yuming Teng, Min Li.

**Writing – original draft:** Mo Yuan.

**Writing – review & editing:** Yuming Teng.

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
