## [Decision Letter · Decision Letter 0]

27 Aug 2025

Dear Dr. Yuan,

Thank you for submitting your manuscript to PLOS ONE. After careful consideration, we feel that it has merit but does not fully meet PLOS ONE’s publication criteria as it currently stands. Therefore, we invite you to submit a revised version of the manuscript that addresses the points raised during the review process.

We look forward to receiving your revised manuscript.

Kind regards,

S Ezhil Vendan, Ph.D

Academic Editor

PLOS ONE

Journal Requirements:

2. Please note that funding information should not appear in the Acknowledgments section or other areas of your manuscript. We will only publish funding information present in the Funding Statement section of the online submission form. Please remove any funding-related text from the manuscript. 

3. Please note that your Data Availability Statement is currently missing the DOI/accession number of each dataset or a direct link to access each database. If your manuscript is accepted for publication, you will be asked to provide these details on a very short timeline. We therefore suggest that you provide this information now, though we will not hold up the peer review process if you are unable.

**Additional Editor Comments:**

The manuscript needs major revision with respect to the reviewers comments. Please add information about the ethical approval for this study.

Reviewers' comments:

Reviewer's Responses to Questions

**Comments to the Author**

1. Is the manuscript technically sound, and do the data support the conclusions?

Reviewer #1: Yes

Reviewer #2: Partly

2. Has the statistical analysis been performed appropriately and rigorously?

Reviewer #1: Yes

Reviewer #2: No

3. Have the authors made all data underlying the findings in their manuscript fully available?

Reviewer #1: Yes

Reviewer #2: Yes

4. Is the manuscript presented in an intelligible fashion and written in standard English?

Reviewer #1: Yes

Reviewer #2: Yes

Reviewer #1: 1).As per the AIM of the study it needs to be a RCT ,however a retrospective analysis with PSM will conclude some findings, but the unmeasured variables like the volume of fluid for resuscitation, the area of infarction and penumbra, neurofunctional scores, type of antibiotics, timing of antibiotics and vasopressors and fluids, will definitely impact PSM hence study findings...2).the paradox in mortality trends are not explainable as albumin addition to sick patients not done on randomization but it was done as the last resort to sick patients ,hence the paradox of having higher mortality in nonearly group followed by early group and least in control group, where as it should be other way that early group should have highest mortality followed by non-early then control group ,hence this statements show that the very sick patients received albumin as resort but not on the basis of randomization, hence skewing the mortality data independently of treatment efficacy. 3) The true temporal impact of albumin administration will be better assessed if A time-segmented mortality analysis—comparing early mortality (e.g., within 7 or 28 days) versus late mortality (beyond 28 days)—could have provided more clinically relevant insights into the direct treatment effects.

Reviewer #2: The manuscript by Yuming Teng et al. presents a retrospective analysis aimed at evaluating whether the administration of crystalloids—the standard first-line therapy for fluid resuscitation in patients with sepsis—supplemented with albumin confers a survival benefit in ischemic stroke patients with sepsis admitted to the ICU, compared with crystalloids alone. Current evidence suggests that albumin use in ischemic stroke is associated with an increased risk of adverse events, including pulmonary edema and intracerebral haemorrhage. On this basis, the authors hypothesize that the addition of albumin to crystalloid therapy is unlikely to improve survival outcomes in this patient population.

A total of 493 patients were included, with 298 receiving crystalloids alone and 195 receiving crystalloids plus albumin. The authors used propensity score matching and multivariable Cox regression analyses to reduce bias. The study found that combination therapy—especially when albumin was administered later than 24 hours (“non-early therapy”)—was associated with significantly higher 90-day mortality and longer ICU and hospital stays, with no consistent survival benefit observed in the subgroups.

Although the study addresses a clinically relevant question in a high-risk patient population, the retrospective design and baseline differences limit the interpretation of the data. Overall, the manuscript provides valuable insights, addressing a major concern in the management of stroke patients with sepsis complications, but major revisions are needed to clarify methodology and data interpretation.

Major Concerns

Methodology

The authors should provide more detail on the use of propensity score matching and specify which variables were included in the multivariable Cox models.

Results and Discussion

1. Confounding by indication: As presented in Table 1, patients receiving albumin were initially sicker, with higher severity scores and more comorbidities. This likely contributes to the observed higher mortality and prolonged ICU stays. The authors should emphasize in the discussion that this may represent an association rather than a causal effect of the treatment.

2. The rationale for the 24-hour cutoff defining early albumin administration is not justified and should be clarified.

3. ICU and hospital length of stay likely reflect disease severity rather than treatment effect; the authors should better redefine in the discussion.

4. Potential adverse events of albumin (pulmonary edema, intracerebral haemorrhage) are hypothesized but were not directly measured in the dataset. The authors should clarify this limitation and ensure that the conclusion emphasizes associations rather than providing direct treatment recommendations

Minor Concerns

• Lines 187–188 are missing the comparison group; please rephrase for clarity.

• The introduction would benefit from a more structured review of the literature on albumin use in both sepsis and ischemic stroke. The authors should contextualize the findings with the literature, distinguishing stroke-focused and sepsis-focused studies.

• Figures 2 and 3 are difficult to read; please improve figure quality and clarity.

**Do you want your identity to be public for this peer review?** For information about this choice, including consent withdrawal, please see our Privacy Policy

Reviewer #1: **Yes: ** Dr. BINGI THRILOK CHANDER

Reviewer #2: **Yes: ** Ester Licastro

---

## [Author Response · Author response to Decision Letter 1]

18 Oct 2025

Dear Editor,

Manuscript number: PONE-D-25-09820

Title: Early combination of albumin with crystalloids administration might not be beneficial for the survival of ischemic stroke patients with sepsis: a retrospective analysis from MIMIC‑IV database

Authors: Yuming Teng, Min Li, Mo Yuan

Thank you for giving us an opportunity to revise our manuscript. We have revised the manuscript carefully according to the reviewers’ comments and recommendations. The response to each question or comment of reviewers was listed point-by-point in the attached pages. We hope that the revised form is suitable for publication in your journal.

Yours sincerely,

Mo Yuan

School of Medicine, Yunnan University, Kunming, China, Dong wai huan South Road, University Town, Chenggong District, Kunming City, Yunnan Province 650000, China.

Tel: 0086-871-65033825

E-mail: ym@ynu.edu.cn

Answers to editor

Major comments

1. Please note that your Data Availability Statement is currently missing the DOI/accession number of each dataset or a direct link to access each database. If your manuscript is accepted for publication, you will be asked to provide these details on a very short timeline. We therefore suggest that you provide this information now, though we will not hold up the peer review process if you are unable.

Answer: Thank you for this important reminder regarding the Data Availability Statement. We have now deposited all relevant datasets in public repositories and updated the Data Availability Statement in our manuscript accordingly. All datasets supporting the conclusions of this study are obtained from the MIMIC-IV database (https://mimic.physionet.org/). The datasets can be obtained from github (https://github.com/neuroym/albumin-combination.git).

Please add information about the ethical approval for this study.

Answer: Thank you for your inquiry. The ethics statement has been relocated to the “Methods” section (page 6) and has been removed from its original location to ensure proper placement. We have updated the Methods section with the following ethics statement: “MIMIC-IV is a publicly available clinical database resulting from a partnership between the Beth Israel Deaconess Medical Center (BIDMC) and the Massachusetts Institute of Technology (MIT). The data originates from routine care at BIDMC and is subsequently deidentified and processed. Access is restricted to investigators who have completed required ethics training and have signed a Data Use Agreement. The Institutional Review Board of BIDMC approved the creation and sharing of this resource under a waiver of informed consent [1] .”

Answers to reviewer 1

Major comments

1. As per the AIM of the study it needs to be a RCT, however a retrospective analysis with PSM will conclude some findings, but the unmeasured variables like the volume of fluid for resuscitation, the area of infarction and penumbra, neurofunctional scores, type of antibiotics, timing of antibiotics and vasopressors and fluids, will definitely impact PSM hence study findings.

Answer: Thank you for your inquiry. We fully acknowledge that these unmeasured variables are crucial in managing ischemic stroke patients with sepsis and could potentially influence the outcomes. In direct response to this concern, we have performed extensive additional analyses to address the impact of these variables, and we have revised the manuscript accordingly.

(1) We have expanded Table 1 to include the following key treatment parameters: timing of vasopressor use, types of antibiotics, timing of antibiotic administration, timing of crystalloid administration, and volume of crystalloid administration. NIHSS scores, along with infarct and penumbra volume, were not available in the MIMIC-IV database, and we have acknowledged this limitation in the “Discussion” section.

(2) To control for these residual confounders, we included them as covariates in our multivariable regression models. After this comprehensive adjustment, the primary findings remained consistent: combination therapy, particularly non-early combination therapy, was consistently associated with an increased risk of 90-day all-cause mortality and prolonged hospital and ICU stays. In addition, our study failed to demonstrate a beneficial effect of early combination therapy on patient outcomes.

(3) We performed subgroup and stratified analyses evaluating the association between combination therapy and 90-day mortality. Importantly, tests for interaction showed no statistical significance for the timing of vasopressor use, timing of antibiotic administration, or timing of crystalloid administration. This suggests that the association between treatment strategy and mortality was not significantly modified by these variables.

(4) We acknowledge that randomized controlled trials (RCTs) are the gold standard. However, the consistent results obtained through our propensity score matching (PSM), multivariate adjustments for key measurable confounders, and extensive subgroup analyses suggest that our primary conclusions are likely to be valid. The findings from these additional analyses have been incorporated into the “Results” and “Discussion” sections of the manuscript.

2. The paradox in mortality trends are not explainable as albumin addition to sick patients not done on randomization but it was done as the last resort to sick patients ,hence the paradox of having higher mortality in nonearly group followed by early group and least in control group, where as it should be other way that early group should have highest mortality followed by non-early then control group ,hence this statements show that the very sick patients received albumin as resort but not on the basis of randomization, hence skewing the mortality data independently of treatment efficacy.

Answer: Thank you for raising this critical point regarding the potential for confounding by indication, wherein albumin administration as a last-resort therapy could skew the mortality data. We acknowledge this as a fundamental limitation inherent to observational studies like ours.

(1) We believe the observed mortality pattern—highest in the non-early group, followed by the early group, and lowest in controls—may not solely be an artifact of patient severity but could also reflect the divergent biological effects of albumin in sepsis versus ischemic stroke. Published literature offers a plausible explanation: early albumin is associated with a mortality benefit in sepsis [2–5], whereas in acute ischemic stroke, it has been linked to increased risks of intracerebral hemorrhage and pulmonary edema and shows no improvement in neurological outcomes [6][7]. In our cohort of patients with both conditions, we postulate a "neutralization-amplification" effect. For the early group, the benefit for sepsis might have been partially neutralized, or even overwhelmed, by the treatment-associated risks for stroke. Conversely, the non-early group suffers from the dual disadvantage of missing the critical treatment window for sepsis and the severe complications in stroke, ultimately leading to the highest mortality.

(2) To address this issue, we employed a variety of methodological approaches. Building on the Propensity Score Matching (PSM) analysis we conducted previously, in response to Reviewer Comment 3, we performed a Time-Segmented Mortality Analysis, and the results actually helps clarify this paradox: it shows that the non-early group's excess mortality occurs exclusively in the late phase (>28 days), with no early survival disadvantage (S6 Table and S7 Table). This temporal pattern suggests that non-early albumin administration, rather than baseline severity alone, drives the poor outcomes. If only the sickest patients received albumin as a last resort, we would expect higher early mortality in both albumin groups, which we did not observe. Furthermore, we explicitly emphasized the associative nature of our study results in the "Discussion" and "Conclusion" section, highlighting the need to establish a causal relationship through randomized controlled trials (RCTs).

(3) In summary, while residual confounding cannot be fully ruled out, the paradoxical trend is supported by a biologically plausible mechanism grounded in existing evidence and the results of a time-dependent analysis. Therefore, we posit that our results, while requiring validation from future RCTs, present a reasonable and clinically relevant finding in this complex patient population.

3. The true temporal impact of albumin administration will be better assessed if A time-segmented mortality analysis—comparing early mortality (e.g., within 7 or 28 days) versus late mortality (beyond 28 days)—could have provided more clinically relevant insights into the direct treatment effects.

Answer: We thank the reviewer for this insightful suggestion. In response, we have conducted a time-segmented mortality analysis. The methodology for this analysis has been detailed in the "Materials and Methods" section, the corresponding results are presented in the "Results" section, and their interpretation has been integrated into the "Discussion". The analysis yielded a distinct temporal pattern. No significant differences in mortality were observed among the groups during the early phase (0-28 days). In contrast, the non-early combination therapy group exhibited a substantially higher risk of mortality in the late phase (HR = 6.18, 95% CI 2.86–13.35, P <0.001), as provided in Supplementary Tables S6 and S7. This temporal dissociation provides critical clinical insights. The absence of an early survival disadvantage argues against baseline patient severity as the sole explanation for the observed outcomes. Instead, it validates the time-dependent nature of the treatment effect and specifically associates non-early albumin administration with significantly worse long-term survival.

Answers to reviewer 2

Major comments

1. The authors should provide more detail on the use of propensity score matching and specify which variables were included in the multivariable Cox models.

Answer: Thank you for your inquiry. We have thoroughly revised the "Statistical Analysis" section to address your specific points regarding Propensity Score Matching (Page 8, Paragraph 2) and the multivariable models (Page 9, Paragraph 3).

2. Confounding by indication: As presented in Table 1, patients receiving albumin were initially sicker, with higher severity scores and more comorbidities. This likely contributes to the observed higher mortality and prolonged ICU stays. The authors should emphasize in the discussion that this may represent an association rather than a causal effect of the treatment.

Answer: We thank the reviewer for raising this important concern. We fully acknowledge that the observed baseline differences in severity scores suggest potential confounding by indication.

(1) To address this, we performed supplementary multivariable Gamma regression model adjusted for key severity indices and comorbidities. These analyses consistently demonstrated that combination therapy—particularly the non-early administration strategy—remained independently associated with significantly prolonged durations of both ICU and hospital stays (Table 2 and S5 Table), thereby substantiating the robustness of our findings.

(2) Regarding mortality, a time-segmented analysis revealed that the elevated mortality in the non-early group was exclusively confined to the late phase (>28 days), with no survival disadvantage observed during the early period (S6 Table and S7 Table). This pattern suggests that the poor outcomes are driven by non-early albumin use rather than baseline severity alone. This interpretation is strengthened by the absence of early survival differences across groups. If higher severity scores and more comorbidities contributes to higher mortality, we would expect higher early mortality in the albumin groups—a pattern that was not observed.

(3) While we employed rigorous methods, including propensity score matching and multivariable regression, to address measured confounders, we explicitly acknowledge that residual confounding by unmeasured factors remains an important limitation. Accordingly, we have emphasized in the “Discussion” that our findings demonstrate an association rather than establish causality.

3. The rationale for the 24-hour cutoff defining early albumin administration is not justified and should be clarified.

Answer: Thank you for your query regarding the rationale for the 24-hour cutoff defining early albumin administration. We agree that this definition requires clear justification based on pathophysiology, clinical evidence, and practical feasibility. Our reasoning is outlined below.

(1) In addition to its superior volume-expanding efficacy and renal safety compared to crystalloids, albumin provides essential biological functions such as maintaining oncotic pressure, antithrombotic activity, and molecular transport. A hallmark of sepsis, endothelial injury, worsens progressively and typically peaks within the first 24 hours [8–10]. This damage exacerbates albumin leakage into the interstitial space, and albumin infusion may unintentionally intensify extravasation[9]�which may exacerbate tissue edema and impair the volume expansion effect. Therefore, the early initiation of albumin within this window, prior to maximal dysfunction of the endothelium, is critical to ensuring its benefits in volume expansion and microcirculatory support are fully realized, before severe leakage compromises its therapeutic effects.

(2) Since 2004, sepsis treatment guidelines have recommended time-based bundles (e.g., 6-h, 3-h, and most recently 1-h) to highlight that "time matters" in sepsis management, and considering feasibility in the intensive care unit, combining albumin with crystalloids within the first 24 hours may be an optimal timeframe[5]. This time frame is logistically feasible in most critical care settings and has been widely adopted in comparable research[2–5,11], facilitating interpretation and cross-study comparison.

We believe this definition is physiologically informed, clinically relevant, and consistent with established research frameworks.

4. ICU and hospital length of stay likely reflect disease severity rather than treatment effect; the authors should better redefine in the discussion.

Answer: We thank the reviewer for this important observation. We agree that the length of stay is profoundly confounded by initial disease severity. To address this concern, we performed a supplementary analysis using a multivariable Gamma regression model adjusted for disease severity scores. The results demonstrated that the association between the treatment strategy and prolonged length of stay remained statistically significant even after accounting for baseline severity (Table 2 and S5 Table). This suggests that ICU and hospital stay may reflect treatment effects beyond those attributable to disease severity. Nevertheless, given the inherent limitations of our retrospective design, we have rephrased the "Discussion" to emphasize the associative nature of these findings and the need for confirmation through prospective studies.

5. Potential adverse events of albumin (pulmonary edema, intracerebral haemorrhage) are hypothesized but were not directly measured in the dataset. The authors should clarify this limitation and ensure that the conclusion emphasizes associations rather than providing direct treatment recommendations.

Answer: We thank the reviewer for this important clarification. We have revised the limitations of the "Discussion" section to explicitly state that pulmonary edema and intracerebral hemorrhage were not directly measured in our dataset, and that these mechanisms are hypothesized based on prior literature rather

---

## [Decision Letter · Decision Letter 1]

11 Nov 2025

Dear Dr.  Yuan,

Thank you for submitting your manuscript to PLOS ONE. After careful consideration, we feel that it has merit but does not fully meet PLOS ONE’s publication criteria as it currently stands. Therefore, we invite you to submit a revised version of the manuscript that addresses the points raised during the review process.

We look forward to receiving your revised manuscript.

Kind regards,

S Ezhil Vendan, Ph.D

Academic Editor

PLOS ONE

Journal Requirements:

Additional Editor Comments:

**The manuscript needs revision.**

The revised manuscript is improved compared to the previous version. Still, the manuscript needs revision with respect to the following comments.

Line 73-75: Authors training course completion detail not required in the manuscript. Remove “To access this database, ………………………………………….. and passed the examination (ID: 57478133).

Please justify with similar studies with respect to the exemption for ethical approval.

Line 89-90: Add reference for the sentence "Access is restricted to investigators who have completed required ethics training and have signed a Data Use Agreement.".

Add figures with good quality.

Reviewer's Responses to Questions

**Comments to the Author**

Reviewer #2: All comments have been addressed

2. Is the manuscript technically sound, and do the data support the conclusions?

Reviewer #2: Yes

3. Has the statistical analysis been performed appropriately and rigorously?

Reviewer #2: I Don't Know

4. Have the authors made all data underlying the findings in their manuscript fully available?

Reviewer #2: Yes

5. Is the manuscript presented in an intelligible fashion and written in standard English?

Reviewer #2: Yes

Reviewer #2: (No Response)

**Do you want your identity to be public for this peer review?** For information about this choice, including consent withdrawal, please see our Privacy Policy

Reviewer #2: No

---

## [Author Response · Author response to Decision Letter 2]

8 Dec 2025

Answers to editor

Major comments

1. Line 73-75: Authors training course completion detail not required in the manuscript. Remove “To access this database, ………………………………………….. and passed the examination (ID: 57478133).

Answer: Thank you for your suggestions. I have deleted the relevant content as per your request.

2. Please justify with similar studies with respect to the exemption for ethical approval.

Answer: Thank you for your strict and rigorous requirements. In accordance with your requirements, we have further justified the rationale for ethical approval exemption

“MIMIC-IV is a public clinical database co-developed by Beth Israel Deaconess Medical Center (BIDMC) and MIT, sourced from BIDMC’s deidentified routine care data. Its creation and public sharing received an ethical waiver from BIDMC’s Institutional Review Board, which granted informed consent exemption and approved data sharing [1]. Access is restricted to investigators who have completed required ethics training and have signed a Data Use Agreement [2]. In compliance with these requirements, our research team has fulfilled all access prerequisites. Therefore, this retrospective study meets the exemption criteria for ethical review, and this practice has an established precedent in peer-reviewed studies utilizing the MIMIC-IV database [3, 4].”

We have supplemented the aforementioned justifications along with citations of References [1], [2], [3], and [4] to the "Ethics Statement" section under the "Methods" part of the manuscript (see Page 6, Lines 84-91 and Page 7, Lines 92-93).

reference

1. Johnson A, Bulgarelli L, Pollard T, Horng S, Celi LA, Mark R. Mimic-iv. PhysioNet; doi:10.13026/6MM1-EK67

2. Johnson AEW, Bulgarelli L, Shen L, Gayles A, Shammout A, Horng S, et al. MIMIC-IV, a freely accessible electronic health record dataset. Sci Data. 2023;10: 1. doi:10.1038/s41597-022-01899-x

3. Zhou S, Zeng Z, Wei H, Sha T, An S. Early combination of albumin with crystalloids administration might be beneficial for the survival of septic patients: A retrospective analysis from MIMIC-IV database. Ann Intensive Care. 2021;11: 42. doi:10.1186/s13613-021-00830-8

4. Li X-Y, Chen W-S, Qu Z-K, Chen J-G, Li L, Li S-N, et al. Early use of albumin may increase the risk of sepsis-associated acute kidney injury in sepsis patients: A target trial emulation. Mil Med Res. 2025;12: 51. doi:10.1186/s40779-025-00641-z

3. Line 89-90: Add reference for the sentence "Access is restricted to investigators who have completed required ethics training and have signed a Data Use Agreement.".

Answer: Thank you for your constructive comments. The relevant references have been added as requested.

4. Add figures with good quality.

Answer: Concerning the figure-related issue, we have repeatedly verified that the submitted images are at a resolution of 600 DPI. Furthermore, when clicking the link "Click here to access/download; Figure; Fig.#-plosone" in the preview PDF, the downloaded images remain extremely clear even when enlarged. Therefore, we kindly request that you verify the matter again. Should any issues arise, please feel free to contact us promptly. Thank you for your assistance.

---

## [Editor Report · Decision Letter 2]

10 Dec 2025

Early combination of albumin with crystalloids administration might not be beneficial for the survival of ischemic stroke patients with sepsis: a retrospective analysis from MIMIC‑IV database

PONE-D-25-09820R2

Dear Dr. Mo Yuan,

We’re pleased to inform you that your manuscript has been judged scientifically suitable for publication and will be formally accepted for publication once it meets all outstanding technical requirements.

Kind regards,

S Ezhil Vendan, Ph.D

Academic Editor

PLOS One
---

## [Editor Report · Acceptance letter]

PONE-D-25-09820R2

PLOS One

Dear Dr. Yuan,

I'm pleased to inform you that your manuscript has been deemed suitable for publication in PLOS One. Congratulations! Your manuscript is now being handed over to our production team.

Kind regards,

on behalf of

Dr. S Ezhil Vendan

Academic Editor

PLOS One